# Do Elite Basketball Players Maintain Peak External Demands throughout the Entire Game?

**DOI:** 10.3390/s24134318

**Published:** 2024-07-03

**Authors:** Hugo Salazar, Filip Ujakovic, Jernej Plesa, Alberto Lorenzo, Enrique Alonso-Pérez-Chao

**Affiliations:** 1Faculty of Education and Sport, University of Basque Country (UPV/EHU), 01007 Vitoria-Gasteiz, Spain; hsalazar002@gmail.com; 2Basketball Club Cedevita Olimpija, 1000 Ljubljana, Slovenia; filip.ujakovic@student.kif.unizg.hr (F.U.);; 3Faculty of Physical Activity and Sport Sciences—INEF, Polytechnic University of Madrid, 28040 Madrid, Spain; 4Department of Physical Activity and Sports Science, University Alfonso X el Sabio, 28691 Villanueva de la Cañada, Spain; ealonsoperezchao@gmail.com; 5Faculty of Sports Science, European University of Madrid, 28670 Villaviciosa de Odón, Spain

**Keywords:** basketball, most demanding scenarios, PlayerLoad, team sport, physical demands, accelerometry

## Abstract

Consideration of workload intensity and peak demands across different periods of basketball games contributes to understanding the external physical requirements of elite basketball players. Therefore, the aim of this study was to investigate the average intensity and peak demands encountered by players throughout game quarters. PlayerLoad per minute and PlayerLoad at three different time samples (30 s, 1 min, and 3 min) were used as workload metrics. A total of 14 professional elite male basketball players were monitored during 30 official games to investigate this. A linear mixed model and Cohen’s d were employed to identify significant differences and quantify the effect sizes among game quarters. The results showed a significant, moderate effect in PlayerLoad per minute between Q1 vs. Q4, and a small effect between Q2 and Q3 vs. Q4. Furthermore, a small to moderate decline was observed in external peak values for PlayerLoad across game quarters. Specifically,, a significant decrease was found for the 3 min time window between Q1 and other quarters. The findings from the present study suggest that professional basketball players tend to experience fatigue or reduced physical output as the game progresses.

## 1. Introduction

Load monitoring has become an essential process for coaches and sports science practitioners to examine the individual workload of players and the collective workload of teams. Additionally, quantifying physical and physiological loads is important for understanding the dose–response nature of the training process when establishing optimal training procedures [1]. Training load includes both external and internal components. External loads (ELs) indicate the physical workload performed (e.g., duration, distance), which is determined by the organization, quality, and quantity of exercise (training plan) [2]. Internal load (IL) refers to the psycho-physiological response during exercise aimed at meeting the demands imposed by the EL (e.g., heart rate, heart rate variability, rate of perceived exertion) [2]. In the modern era of sports science, recent technological advancements, such as electronic performance tracking systems (EPTSs) integrating inertial measurement units (IMUs) like accelerometers and gyroscopes, have transformed the monitoring of basketball players in both training sessions and actual games [3].

Recent studies utilizing EPTSs have characterized basketball as an intermittent, high-intensity sport. These studies have shown that the majority of playing time (93.65%) is spent in standing–walking (<7 km/h) and jogging activities (7–14 km/h) [4]. These periods of low-motion activities are interspersed with the most demanding scenarios (e.g., peak demands, high-intensity period). During these moments, players constantly engage in continuous changes in direction, jumps, accelerations, decelerations, physical contacts, and specific basketball skills (e.g., crossovers, lay-ups) [5,6]. Investigating the physical demands experienced by athletes during both competition and training has emerged as a focal point in sports science research [7]. This expanding field enables sports coaches to gather precise data for refining athletic training programs to target specific adaptations. To accomplish this, it is crucial to implement training methods that not only capture average values but also replicate the challenges posed by peak competition demands, thereby optimizing overall athletic performance [6].

Peak demands (PDs) refer to the most intense activity experienced by players within a specified timeframe for a chosen variable [6]. PDs in basketball players have been assessed using various external variables (e.g., PlayerLoad or distance) and time windows (e.g., 30 s, 45 s, 1 min, 5 min, and 10 min) [6,8,9,10]. The literature on PDs in basketball has examined the influence of contextual factors such as player position [6,11], game score-line [12,13], game schedule [14], team venue [15], cumulative playing time across the entire game [9,16] and prior to intense passages [9], activity type [17,18], age category [19] or moment of the game [10,20]. However, these studies are often conducted on non-professional samples, primarily due to the challenges associated with accessing professional players.

Previous research on professional basketball has examined the impact of various factors on the external PDs experienced by players, including playing positions [21,22] and the seasonal period [23]. There is a dearth of studies in the literature that assess professional or elite basketball players, potentially due to the challenges associated with accessing data within high-performance environments or the limited availability of microtechnology for quantifying competition demands in certain professional leagues [24]. Furthermore, the imprecise usage of terms such as “elite”, “high performance”, and “professional” in sports contexts introduces ambiguity and may result in research spanning various competitive levels being classified as “elite”. This lack of specificity complicates the interpretation and comparison of findings across studies [25,26].

Understanding the fluctuations in PDs provides a deeper insight into the demands of competition. Fluctuations refer to variations in the intensity, measurement, or quality of a variable over a period of time [20]. One of the earlier published articles evaluating fluctuations throughout the quarters of a game demonstrated how dribbling actions and total activity velocities declined as the game progressed [27]. Furthermore, with the implementation of microtechnology during games and among professional second-division players, it has been revealed that all average external physical demands decreased across game quarters [21]. Expanding the scope to include different levels of play, research conducted among semi-professional [10,22] or youth basketball players [20] has uncovered a similar trend in peak demands as observed in average demands, with higher peak values typically recorded in the first quarter compared to the latter stages of the game [10,20,22]. Collectively, these findings underscore the intricate dynamics of physical performance throughout a basketball game, influenced by factors such as player expertise, match intensity, and strategic maneuvers employed by the teams [10,20,22].

Based on the preceding information, our understanding of the behavior of external peak demand during official basketball matches among elite or professional players remains limited. Consequently, it cannot be assumed that the differences in external PDs observed between quarters can be generalized to other samples (e.g., female players or professional players) [9]. Furthermore, separate research on this topic is necessary for elite male players, as top male basketball players have been shown to face significantly higher physical demands and strategic challenges during games compared to their younger or semi-professional counterparts [28]. Understanding these fluctuations in external peak requirements could offer several advantages to basketball practitioners: (1) the ability to develop more precise conditioning regimens, (2) the optimization of player rotations during games to maintain optimal physical performance throughout game periods, (3) the development of strategies for prescribing training drills more accurately based on real game reference values, and (4) preparing players to sustain the physical demands of a basketball game when returning from injury. Therefore, the aim of the present study was to investigate the average PDs encountered by professional basketball players across game quarters within three distinct timeframes (30 s, 1 min, and 3 min). Based on prior research conducted with non-elite male players utilizing microtechnology, it was hypothesized that the average [21] and peak values [10,20,22] would likely decrease across the quarters throughout the game.

## 2. Materials and Methods

### 2.1. Participants

Professional elite male basketball players classified as Tier 4 (elite/international level) [25,26] (n = 14, mean ± standard deviation: age: 27.8 ± 3.5 y.0; height: 198.1 ± 10.4 cm; body mass: 97.4 ± 11.6 kg) were monitored during 30 games (30 Eurocup/ABA league). Throughout the data collection period, players engaged in an average of 10 h of training per week, comprising 5 basketball sessions and 2 resistance training sessions. Additionally, they played 2 games weekly. Players included in the study were from all positions: guards, forwards, and centers. The team included in the study is based in Slovenia and has played in the ABA League and Eurocup. Eurocup fixtures were scheduled between Tuesday and Friday, while domestic league games were typically held on weekends.

Game samples from each player were only retained in the final analysis if they completed a minimum of 4 min of playing time derived from devices on each quarter. Playing time derived from devices included all stoppages in play, such as free-throws, fouls, and out-of-bounds, but excluded warm-ups, break periods between quarters, time-outs, or time when players were substituted out of the game [9]. Samples obtained from game instances where players accrued less than 4 min of playing time, as determined by data derived from the monitoring devices, were systematically excluded from the final analyses. This exclusion criterion ensured that only substantial periods of active participation were considered, thus minimizing the impact of brief or negligible contributions to the overall PlayerLoad assessment. Additionally, any player who prematurely exited the match due to injury or experienced a cessation of device functionality, whether due to battery depletion or technical malfunctions, was automatically excluded from the dataset corresponding to that specific match. This meticulous approach to data curation aimed to uphold the integrity and reliability of the analytical outcomes by focusing exclusively on instances where players were actively engaged in gameplay for a meaningful duration. In this regard, all players had to compete in at least 50% of the total monitored matches. Overall, 958 game samples across the 14 players were included in the analyses. All subjects gave their informed consent for inclusion before they participated in the study. The study was conducted in accordance with the Declaration of Helsinki, and the protocol was approved by the Ethics Committee of the University of Pais Vasco (UPV/EHU, code M10_2018_027).

### 2.2. Design

The study was conducted during the 2023–2024 season. Each player wore a monitoring device (S7, Catapult Sports, Melbourne, Australia) inserted into a fitted neoprene vest underneath their regular playing attire. The device was positioned on the upper thoracic spine between the scapulae [29]. Each device contained microsensor technology consisting of an accelerometer (±16 g, 100 Hz), magnetometer (±4.900 µT, 100 Hz), and gyroscope (up to 2000 deg/s, 100 Hz). All players participating in the study were already acquainted with the monitoring technology, having utilized similar devices extensively during both training sessions and competitive games throughout the preceding season. This familiarity ensured a smooth transition into the data collection phase, as players were accustomed to wearing the devices and understanding their functionalities. To maintain consistency and reliability in the recorded data, the devices were activated approximately 20 to 40 min before the commencement of the warm-up phase preceding each game. By initializing the devices ahead of time, any necessary calibration or synchronization processes could be completed without impinging on the players’ pre-match routines. Furthermore, to minimize potential discrepancies arising from variations between the individual units of the monitoring devices, players were assigned the same device for the entirety of the study period. In this regard, the same individual took charge of editing all monitored sessions to minimize inter-rater error to its lowest possible extent. This approach effectively mitigated the risk of inter-unit variability in output readings, ensuring that the data obtained remained consistent and comparable across all participants [30].

### 2.3. Variables

PDs were calculated for PlayerLoad™ (PL) in absolute values and extracted for each player and time window (30 s, 1 min, and 3 min). Research has identified these sample durations as the most practical for consideration in basketball [31]. Average demands were extracted as relative values (Pl·min). PL, a parameter commonly used to measure external load in various sports [6,32,33,34], is derived from accelerometer data and captures the athlete’s accelerations in different planes. However, its definition may vary depending on the manufacturer of the accelerometer device. Recent studies have incorporated PL and have demonstrated its strong correlation with physical and physiological performance [14,35]. Specifically, in the case of basketball, it has the potential to provide a good estimate of the external load of the athlete, as it is a sport that involves a high number of accelerations and decelerations, changes in direction, or explosive efforts. PL was calculated as the square root of the sum of the instantaneous rate of change in acceleration in the three planes of movement (x, y, and z axes) using the following formula [13]: $PlayerLoad™=\frac{\left[\sqrt{{\left({fwd}_{t=i+1}-{fwd}_{t=i}\right)}^{2}}+\sqrt{{\left({side}_{t=i+1}-{side}_{t=i}\right)}^{2}}+\sqrt{{\left({up}_{t=i+1}-{up}_{t=i}\right)}^{2}}\right]}{100}$, where “fwd” indicates movement in the anteroposterior direction, “side” indicates movement in the medial–lateral direction, “up” indicates vertical movement, and t represents the time.

The average and PDs were determined directly from the Catapult software (OpenField v8, Catapult Innovations, Melbourne, Australia). Peak values were calculated as rolling averages, which is a more precise technique for measuring PD compared to fixed methods [36,37] and has been previously used in basketball research [6,8,31]. After extraction, PDs were input into customized Microsoft Excel (version 16.0, Microsoft Corporation, Redmond, WA, USA) spreadsheets for further analysis.

### 2.4. Statistical Analysis

The mean and standard deviation (SD) were determined for PL variables across each sample duration (30 s, 1 min, and 3 min). Data distribution normality and sphericity were validated through the Shapiro–Wilk statistic and Levene’s Test for homogeneity of variances. A linear mixed model (LMM) with Bonferroni post hoc tests considering significance at *p* < 0.05 was used to compare peak values for each sample duration. Game quarter (4 levels) was entered as a fixed factor, while player (n = 14) was entered as the random term. To identify the magnitude of the differences between quarters, effect sizes (ESs) (Cohen’s d) with 95% confidence intervals were calculated. The ES magnitudes of the differences were interpreted as follows: ≤0.2, trivial; >0.2, small; >0.6, moderate; >1.2, large; >2.0, very large; and >4.0, nearly perfect [38]. All statistical analyses were conducted using the software jamovi 2.3 (the jamovi project, 2022) for Windows.

## 3. Results

PL·min across game quarters (Qs) is presented in Figure 1. A decreasing pattern for average values was found among quarters with significant differences (*p* < 0.001) from the first three quarters compared to the last (Q1 vs. Q4, moderate; Q2 vs. Q4, small; Q3 vs. Q4, small).

Table 1 presents the descriptive values of PL for the three sample durations across game quarters and the entire game. Table 2 illustrates the pairwise comparison of PL at different time epochs. For the 30 s sample duration, a significant difference with a *small* effect was observed between game quarters. Regarding the 1 min sample, differences in PL between quarters were significant between Q1 versus Q3 and Q4 (*p* < 0.001, *small* to *moderate*), Q2 with Q3 (*p* < 0.001, *moderate*), and Q3 with Q4 (*p* < 0.05, *small*). For the 3 min sample duration, there was a significant decline in PL from the first three quarters compared to Q4, with a *small* to *moderate* effect. The only non-significant comparison for the 3 min sample duration was between Q2 and Q3.

## 4. Discussion

The aim of the present study was to investigate the average physical demands and PDs encountered by professional basketball players across game quarters within three distinct timeframes (30 s, 1 min, and 3 min). This study provides impactful findings for basketball coaches and performance staff, demonstrating a clear decrease in both average and peak external load values for PL across game quarters. Significant differences were observed between the first three quarters and the last quarter, indicating a decline in player activity as the game progressed.

When contrasting our findings with the existing basketball literature, we found similarities with García et al. (2020), who examined male basketball players competing in the Spanish Second Division. They observed a decrease in average physical demands from the first to the fourth quarters, specifically in total distance covered (*p* < 0.001; ES = −1.31) and PL (*p* < 0.001; ES = −1.27) [21]. Regarding peak values, the current research further investigated external PDs across quarters in basketball, focusing on elite junior male players [20], semi-professional male players (Fox et al., 2021), and male basketball players competing in the Spanish Second Division [22]. Their studies also identified significant decreases throughout the game, with the most notable declines in external peak values occurring between the first and fourth quarters for total distance [20,22], PL [10,20,22] and high-speed running [20,22]. However, the differences between quarters in jogging, running, acceleration, and deceleration for all sample durations (30 s and 45 s and 1, 2, and 5 min) were non-significant [20]. These results highlight the general trend of declining average and peak physical demands as the game progresses, indicating that this decrease is independent of the level of competition.

The hypothesis supporting this phenomenon (decline in average and peak values as the game progresses) may be attributed to fatigue-related mechanisms associated with accumulated playing time throughout entire games and prior to intense periods [9]. It may also depend on situational variables such as the team lineup, which can vary due to differences in player capacities, team cohesion, and tactical approaches, as well as the stage of the game (e.g., the game pace may decline during latter periods) [9,39]. Longer sample durations were more sensitive for detecting differences in PDs between quarters, indicating their usefulness in planning game-like conditioning. In contrast, shorter sample durations may reflect situational load demands during live play. This suggests that longer-duration samples can provide insights into conditioning, while shorter-duration intervals can help understand the demands of live play. Overall, these findings underscore the importance of considering both the duration of sampling intervals and the specific quarters of play when analyzing PL dynamics.

### 4.1. Practical Applications

Our findings can offer a useful practical application for basketball practitioners in several ways. First, training programs can be designed to enhance players’ endurance and recovery capacity to sustain performance throughout the game. For instance, training regimens can incorporate drills with elevated PL towards the end of the session, thereby aiming to bolster conditioning levels during periods away from competitive fixtures. Conversely, an alternative approach could involve introducing tasks with diminished PL towards the session’s conclusion, more accurately simulating the conditions typically encountered during match play. This approach focuses players on cognitive or tactical aspects, which are often crucial late in the game. The goal is for players to work on strategic aspects in situations of fatigue.

Another application could be strategically rotating players during the game or practice to distribute playing time more equitably and maintain optimal performance. This approach would likely involve pre-match or pre-training planning by coaching staff regarding player participation and rest times. By scheduling strategic rest intervals, coaches can mitigate fatigue and optimize player performance at critical moments.

Finally, it is advisable to use real-time performance tracking technologies to monitor players’ physical condition and performance during matches and training sessions. This enables adjustments to participation times based on their physical condition. Overall, understanding how physical demands vary throughout the game can help optimize training, player rotation, and game strategies to maximize team performance across all quarters.

### 4.2. Limitations

The research conducted presents important strengths, such as the inclusion of elite and professional players, which highlights the high level of the sample and the large sample of matches in international-level competition. Nevertheless, some limitations should be considered when interpreting the current findings. First, variables such as score-line, playing position, fixture congestion, players’ role, quality of opposition, tactical aspects, and other factors that could have directly or indirectly influenced the results were not controlled for. Therefore, to advance our understanding of average and peak fluctuations, future research should investigate these factors.

Second, another limitation is the focus on a single variable (PL) due to the challenges associated with installing a local positioning system in elite stadiums, which would allow for measuring positional variables such as distance. Future research conducted with elite or professional players should consider this aspect.

Additionally, a significant limitation is the lack of studies on unexamined populations, such as referees or female players. Research analyzing physical fluctuations has predominantly focused on elite male basketball players [10,20,22]. Incorporating other populations, such as referees who also play a crucial role in the game or female athletes whose playing style and physical demands may differ, could provide a more comprehensive perspective on average and peak fluctuations in sports performance. Therefore, future research should include these populations for a more holistic understanding of the topic.

## 5. Conclusions

The results of this study show a consistent decrease in both the mean and peak values of external physical demands throughout the quarters of a game in professional basketball players, with the most notable reductions occurring between the initial three quarters and the last quarter. These results suggest that professional basketball players tend to experience fatigue or a reduction in physical demands as the game unfolds, underscoring the importance of managing the players’ workload and implementing appropriate strategies to optimize performance throughout the game. Another possible explanation could be that tactical aspects related to game management (such as rotations, playing styles, and defense strategies) during the last quarter influence these physical demands.

## Figures and Tables

**Figure 1 sensors-24-04318-f001:**
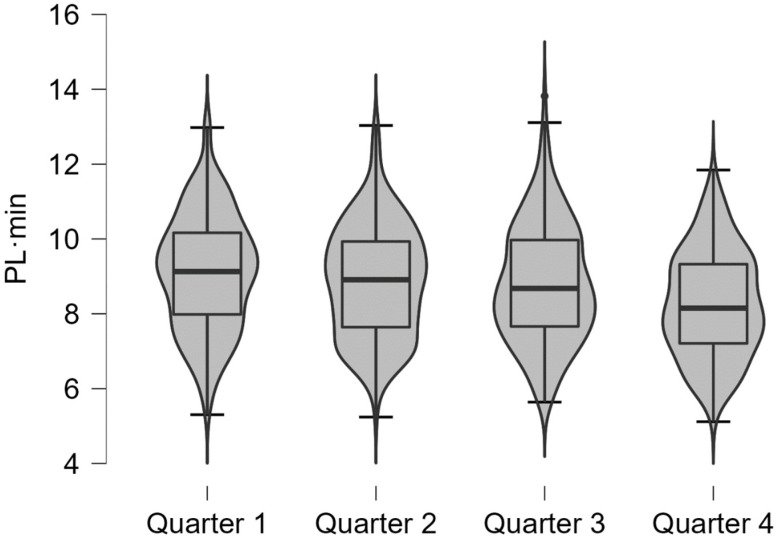
Average values (PL·min) across basketball quarters for elite basketball players.

**Table 1 sensors-24-04318-t001:** Descriptive values (mean ± standard deviation) in PL between game quarters for each sample duration.

Game Quarters	PL 30 s (AU)	PL 1 min (AU)	PL 3 min (AU)
Q1	10.9 ± 1.6	17.4 ± 3	37.1 ± 6.6
Q2	10.8 ± 1.6	16.9 ± 2.6	35.0 ± 6
Q3	10.4 ± 1.6	16.5 ± 2.8	34.9 ± 6.2
Q4	10.1 ± 1.6	15.8 ± 2.5	33.4 ± 5.5

*Note*: Q: quarter; PL: PlayerLoad.

**Table 2 sensors-24-04318-t002:** Pairwise comparison of PL across the three selected timeframes for each quarter of basketball games.

Sample Duration	Effect Size	95% CI	*p*
**30 s sample**			
Q1 vs. Q2	0.07	(−0.17–0.31)	1.00
Q1 vs. Q3	0.30	(0.05–0.55)	<0.05
Q1 vs. Q4	0.47	(0.22–0.72)	<0.001
Q2 vs. Q3	0.23	(−0.01–0.47)	0.073
Q2 vs. Q4	0.40	(0.16–0.64)	<0.001
Q3 vs. Q4	0.17	(−0.08–0.42)	0.44
**1 min sample**			
Q1 vs. Q2	0.17	(−0.07–0.41)	0.37
Q1 vs. Q3	0.36	(0.10–0.60)	<0.001
Q1 vs. Q4	0.57	(0.32–0.82)	<0.001
Q2 vs. Q3	0.18	(−0.06–0.42)	0.30
Q2 vs. Q4	0.40	(0.15–0.64)	<0.001
Q3 vs. Q4	0.22	(0.03–0.47)	<0.05
**3 min sample**			
Q1 vs. Q2	0.35	(0.10–0.60)	<0.001
Q1 vs. Q3	0.36	(0.10–0.62)	<0.001
Q1 vs. Q4	0.62	(0.36–0.87)	<0.001
Q2 vs. Q3	0.01	(−0.24–0.26)	1.0
Q2 vs. Q4	0.27	0.02–0.52	<0.05
Q3 vs. Q4	0.26	(0.01–0.51)	<0.05

*Note*: Q: quarter; CI: confident interval; *p*: *p* value.

## Data Availability

The data are not available due to privacy or ethical restrictions.

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
