# Peer review of "Do Elite Basketball Players Maintain Peak External Demands throughout the Entire Game?"

_sensors, 2024, doi:10.3390/s24134318_

Round 1
Reviewer 1 Report
Comments and Suggestions for Authors
The writing of the paper is not standardized, and it is not a scientific paper. I suggest rejecting this paper. The existing problems are as follows:
1. The similarity of the paper is a bit high, it is best to reduce it to 20% or less.
2. The number of words in the paper does not meet the requirements of Journal.
3. The citation order of references is chaotic, and the summary of references is not detailed enough.
4.There are no innovative methods or models in the paper.
5. In 2.4, why are the sample duration (30-s,1-min,3-min) selected? If the sample duration are larger or lower, what will the result be?
6. In 3, P<0.001, How is this value determined? You should give the reason.
7. Magnitudes of the differences were defined as follows………., What is the basis for the definition of these values?
Comments on the Quality of English Language
Extensive editing of English language required.
Author Response
REVIEWER 1
The writing of the paper is not standardized, and it is not a scientific paper. I suggest rejecting this paper. The existing problems are as follows:
Response: Thank you very much for the feedback. We have meticulously reviewed the manuscript, striving to maintain consistency throughout. Additionally, we have implemented several improvements to ensure the content is clear and cohesive. We appreciate your time and consideration in reviewing our work.
- The similarity of the paper is a bit high; it is best to reduce it to 20% or less.
Response: Thank you for pointing this out. We will carefully review the manuscript to identify and address the sections with high similarity. Our goal is to ensure that the similarity index is reduced to 20% or less. We appreciate your feedback and will take the necessary steps to improve the originality of our work.
- The number of words in the paper does not meet the requirements of Journal.
Response: Thank you for your feedback. Following the reviewers' comments, we have expanded the manuscript, and it currently stands at approximately 3,400 words, excluding references and the abstract. We believe that the article is comprehensive and detailed, covering all relevant aspects of the study. However, we understand the journal's requirement of reaching 4,000 words, and we are willing to make the necessary adjustments to meet this requirement if it is mandatory. We appreciate your guidance and hope the revised version meets the necessary standards.
- The citation order of references is chaotic, and the summary of references is not detailed enough.
Response: Thank you for your feedback. According to the journal's guidelines, references should be listed in the order of citation rather than alphabetically. Could you please specify what aspects of the citation order you found chaotic? Regarding the details, we have included the journal names in some citations. Additionally, we have thoroughly reviewed all references to ensure they comply with the journal's guidelines.
- There are no innovative methods or models in the paper.
Response: Thank you very much for your comment. We are open to any specific suggestions for new methods and models that could be included to improve the article while maintaining its objective. We would like to highlight that the innovation of our paper lies in the inclusion of an entire season of professional players. Monitoring these matches is extremely challenging, as most leagues do not permit such comprehensive data collection.
- In 2.4, why are the sample duration (30-s,1-min,3-min) selected? If the sample duration are larger or lower, what will the result be?
Response: Thank you for your question. Short windows (15, 30, 45 seconds, and 1 minute) provide information on peak values related to the nature of the sport, as actions in basketball are generally intermittent and typically do not exceed one consecutive minute in duration. Longer windows (2, 3, 4, 5 minutes) allow basketball practitioners to compare peak demand with practice drills, which, depending on the coach's approach, usually last within this duration without stoppages. Indeed, 30-second, 1-minute, 2-minute, and 5-minute windows have been identified as the most practical to consider in basketball.
In this regard, we have selected three durations to avoid making the results overly extensive: a short window (30 seconds), a 1-minute window to facilitate relative comparisons with other studies, and a long window (3 minutes), as these are commonly used durations. Additionally, we have included relevant references to support this choice as follows: Research has identified these sample durations as the most practical for consideration in basketball [33]. (Line 164-165)
- In 3, P<0.001, How is this value determined? You should give the reason.
Response: Thank you for the details. These p-values are the statistical results from the linear mixed model analysis conducted to compare the average values across quarters. Since the p-value is less than 0.05, we can conclude that there are significant differences among the quarters
.7. Magnitudes of the differences were defined as follows………., What is the basis for the definition of these values?
Response: Thank you. The basis for the definition of these values for interpreting effect sizes (Cohen's d) comes from the need to provide a standardized way to understand and communicate the magnitude of differences observed in research studies. These values help to quantify the practical significance of results, offering more context than mere statistical significance. In this regard, we have made a small change to the sentence to make it less confusing.
Reviewer 2 Report
Comments and Suggestions for Authors
The authors wrote an interesting manuscript about the external load during the dozen games of elite male basketball players. The expertise of the authors is evident from the text.
Introduction
The "Introduction" is relevant to the topic and generally provides a quality background for the research. The study's aim is clear.
Line 47: Non-standard abbreviation for kilometers per hour.
Line 98: The study of Abdelkrim et al. (2010) is not cited correctly.
Materials and Methods
Part "Materials and Methods" is comprehensible. However, some information about the participants is missing. The authors should add information about the regular training regime during the week (sessions per week) and the players' positions. The authors should also state the country where the players/team played for better comparison in further studies.
The authors should provide information regarding the ethical aspects of the study. While I am unsure of the legislative requirements in the country where the research was conducted, obtaining informed consent from participants before beginning the research is generally standard practice. Additionally, the study should have been approved by the institutional or local ethics committee. Furthermore, it should be noted that the study adhered to the principles of the Declaration of Helsinki.
Lines 163–164: The sentence "positive results at the time to record." does not make sense in this context.
Results, Discussion, and Conclusions
Results are presented adequately. The Discussion and Conclusions are conducted within the results' framework. I appreciate the study's limitations, where the authors know the limits and future research direction. In future studies, the analysis should be more concerned with the contextual factors.
References
References no. 4, 9, 18, 19, 20, 23, 24, and 31 are not cited correctly – the journal's name is missing.
In conclusion, the manuscript is well-written. Except for a few shortcomings, this is a quality paper. I recommend publishing the article after minor corrections.
Author Response
REVIEWER 2
Introduction
The "Introduction" is relevant to the topic and generally provides a quality background for the research. The study's aim is clear.
Response: Thank you for your feedback! We appreciate your positive comments on the relevance of the introduction and clarity of the study's aim.
Line 47: Non-standard abbreviation for kilometers per hour.
Response: Thank you for bringing this to our attention. We have modified the abbreviation to adhere to standard conventions. (Line 45)
Line 98: The study of Abdelkrim et al. (2010) is not cited correctly.
Response: We apologize for the error and have corrected the citation of the study by Abdelkrim et al. (2010). Thank you.
Materials and Methods
Part "Materials and Methods" is comprehensible. However, some information about the participants is missing. The authors should add information about the regular training regime during the week (sessions per week) and the players' positions. The authors should also state the country where the players/team played for better comparison in further studies.
Response: Thank you for your feedback. We have included the missing information as follow: “Throughout the data collection period, players engaged in an average of 10 hours of training per week, comprising 5 basketball sessions and 2 resistance training sessions. Additionally, they played 2 games weekly. Players included in the study were from all positions: guards, forwards, and centers. The team included in the study is based in Slovenia and played in the ABA League and Eurocup. Eurocup fixtures were scheduled between Tuesday and Friday, while domestic league games were typically held on weekends”. (Line 114-119)
The authors should provide information regarding the ethical aspects of the study. While I am unsure of the legislative requirements in the country where the research was conducted, obtaining informed consent from participants before beginning the research is generally standard practice. Additionally, the study should have been approved by the institutional or local ethics committee. Furthermore, it should be noted that the study adhered to the principles of the Declaration of Helsinki.
Response: We apologize for the oversight. The ethical aspects of the study have been addressed as suggested. “All subjects gave their informed consent for inclusion before they participated in the study. The study was conducted in accordance with the Declaration of Helsinki, and the protocol was approved by the Ethics Committee of University of Pais Vasco (UPV/EHU, code M10_2018_027)”. (Line 136-140)
Lines 163–164: The sentence "positive results at the time to record." does not make sense in this context.
Response: We have removed the sentence as it doesn't make sense, as you rightly pointed out.
Results, Discussion, and Conclusions
Results are presented adequately. The Discussion and Conclusions are conducted within the results' framework. I appreciate the study's limitations, where the authors know the limits and future research direction. In future studies, the analysis should be more concerned with the contextual factors.
Response: Thank you for your feedback. We appreciate your acknowledgment of the adequacy of the results presentation and the alignment of the discussion and conclusions within the framework of the results.
References
References no. 4, 9, 18, 19, 20, 23, 24, and 31 are not cited correctly – the journal's name is missing.
Response: Thank you. We have included the journal names in citations no. 4, 9, 18, 19, 20, 23, 24, and 31. Additionally, we have thoroughly reviewed all references to ensure they comply with the journal's guidelines.
In conclusion, the manuscript is well-written. Except for a few shortcomings, this is a quality paper. I recommend publishing the article after minor corrections.
Response: Thank you very much for the comment. Following the feedback from other reviewers, the minor revisions mentioned have been improved as well as all the English throughout the manuscript.
Round 2
Reviewer 1 Report
Comments and Suggestions for Authors
The paper has been revised according to my suggestions,and I suggest that the paper can be accepted.